# Instance-wise Batch Label Restoration via Gradients in Federated Learning

**Kailang Ma,**[*] **Yu Sun**[*][†] **Jian Cui, Dawei Li, Zhenyu Guan, Jianwei Liu**
School of Cyber Science and Technology, Beihang University, China
`{makailang,sunyv,cuijianw,lidawei, guanzhenyu,liujianwei}@buaa.edu`

## Abstract

Gradient inversion attacks have posed a serious threat to the privacy of federated learning. The attacks search for the optimal pair of input and label best matching the shared gradients and the search space of the attacks can be reduced by pre-restoring labels. Recently, label restoration technique allows for the extraction of labels from gradients analytically, but even the state-of-the-art remains limited to identify the presence of categories (i.e., the class-wise label restoration). This work considers the more real-world settings, where there are multiple instances of each class in a training batch. An analytic method is proposed to perform instance-wise batch label restoration from only the gradient of the final layer. On the basis of the approximate recovered class-wise embeddings and post-softmax probabilities, we establish linear equations of the gradients, probabilities and labels to derive the *Number of Instances* (*NoI*) per class by the *Moore-Penrose pseudoinverse* algorithm. Untrained models are most vulnerable to the proposed attack, and therefore serve as the primary experimental setup. Our experimental evaluations reach over 99% *Label existence Accuracy* (*LeAcc*) and exceed 96% *Label number Accuracy* (*LnAcc*) in most cases on three image datasets and four untrained classification models. The two metrics are used to evaluate class-wise and instance-wise label restoration accuracy, respectively. And the recovery is made feasible even with a batch size of 4096 and partially negative activations (e.g., Leaky ReLU and Swish). Furthermore, we demonstrate that our method facilitates the existing gradient inversion attacks by exploiting the recovered labels, with an increase of 6-7 in PSNR on both MNIST and CIFAR100. Our code is available at https://github.com/BUAA-CST/iLRG.

## 1 Introduction

Federated Learning (FL) is one of the most popular distributed learning paradigms to achieve privacy preservation and has attracted widespread attention (Jochems et al., 2016; McMahan et al., 2017; Yang et al., 2019a), especially in privacy-sensitive fields such as healthcare (Brisimi et al., 2018; Sadilek et al., 2021) and finance (Yang et al., 2019b; Long et al., 2020). FL requires participants to communicate the gradients or weight updates instead of private data on a central server, in principle, offering sufficient privacy protection.

Contrary to prior belief, recent works have demonstrated that shared gradients can still leak sensitive information in FL. Multiple attack strategies preliminarily look into the issue despite their own limitations. For instance, Membership Inference (Shokri et al., 2017) allows the adversary to determine whether an existing data sample is involved in the training set. Property Inference (Melis et al., 2019) analogously retrieves certain attributes (e.g., people's race and gender in the training set). Model Inversion (Fredrikson et al., 2015) utilizes the GAN (Goodfellow et al., 2014) model to generate visual alternatives that look similar but are not the original data. An emerging research, i.e., Deep Leakage from Gradients (DLG) (Zhu et al., 2019), has showed the possibility of fully recovering input data given gradients in a process now known as *gradient inversion*. This approach primarily rely on the *gradient matching* objective to perform the attack, i.e., optimizing a dummy

---

[*]Co-first authors.
[†]Corresponding author.

input by minimizing the distance between its gradient to a target gradient sent from a client. Furthermore, it enables pixel-wise accurate reconstruction on image classification tasks, and soon scales to deeper networks and larger-resolution images in a mini-batch (Geiping et al., 2020; Yin et al., 2021; Jeon et al., 2021; Li et al., 2022).

Gradient inversion jointly recovers private inputs and labels, whereas most works focus more on restoration of training samples. Label restoration is non-trivial, and it has been shown to be critical to high-quality data reconstruction. The optimization-based method for label recovery is not guaranteed to succeed as well. Analytical extraction of the ground truth label from gradients produced by a single sample (Zhao et al., 2020) was first made possible through an innovative observation of the gradient sign. Along this line, follow-ups (Yin et al., 2021; Dang et al., 2021) extend this method to the recovery of labels in a mini-batch with a high success rate. Despite remarkable progress, such attacks are applicable only when no two inputs in the batch belong to the same class. In real-world scenarios, there are multiple instances of each category, which means existing attacks can only determine which classes of samples are present at best. Moreover, most prior methods assume that the target model employs a non-negative activation function, and they fail to handle activation functions that might produce negative elements (e.g., Leaky ReLU (Maas et al., 2013) and Swish (Ramachandran et al., 2017) in EfficientNet (Tan & Le, 2019)).

To this end, our work aims to identify the ground-truth label for each data point in one batch (i.e., instance-wise labels). Firstly, we are able to recover class-wise averaged inputs (i.e., embeddings) to the final layer from its gradients with relative fidelity. The same approach can be proved perfectly correct in the single-sample case (Geiping et al., 2020), but here we require two empirical properties (Sun et al., 2021) to make three approximations: *intra-class uniformity and concentration of embedding distribution* and *inter-class low entanglement of gradient contributions*. Then we derive that the gradient w.r.t. the network final logits equals the difference between the post-softmax probability and the binary representation of the label, and expand it to a system of equations consisting of the gradients, probabilities and labels, where all three variables can be substituted or decomposed equivalently. We finally obtain a *Least Square Solution* (*LSS*) of the *Number of Instances* (*NoI*) per class by the *Moore-Penrose pseudoinverse* algorithm.

Our main contributions are as follows:

- We propose an analytic method to recover the exact labels that a client possesses via batch-averaged gradients in FL. Following the approximate restoration of class-wise embeddings and post-softmax probabilities, a system of linear equations with the gradients, probabilities and labels is established to derive the NoI of each class by the Moore-Penrose pseudoinverse algorithm.

- Our method outperforms prior attacks and poses a greater threat to the untrained model. In this case, it works on large batch sizes of up to 4096, and handles classification models with partially negative activations as deep as ResNet-152 (He et al., 2016).

- We demonstrate that gradient inversion attack can be combined with our batch label restoration to further improve its performance due to the reduced search space during optimization.

## 2 PREMILINARIES AND RELATED WORK

### 2.1 PROBLEM FORMULATION

Given a network with weights $\mathbf{W}$ and the batch-averaged gradient $\nabla \mathbf{W}$ calculated from a batch of sample-label pairs, we expect to reveal instance-wise labels $\mathbf{y}$ via gradients. For each pair $(\mathbf{x}, \mathbf{y})$, we denote the embedding vector into the final layer as $\mathbf{e} \in \mathbb{R}^m$, the network final logits as $\mathbf{z} \in \mathbb{R}^C$ and the post-softmax probability as $\mathbf{p} \in \mathbb{R}^C$ in range $(0, 1)$, where m is the embedding dimension, $C$ is the number of classes and $\mathbf{y}$ here is the one-hot binary representation of the same shape as $\mathbf{z}$. In the following sections, $\mathbf{W} \in \mathbb{R}^{C \times m}$ and $\mathbf{b} \in \mathbb{R}^C$ refer to the weight and bias of the final classification layer, respectively. Then we have $\mathbf{z} = \mathbf{We} + \mathbf{b}$ and $\mathbf{p} = SoftMax(\mathbf{z})$.

**Threat Model**: As prior works (Zhu et al., 2019; Geiping et al., 2020; Yin et al., 2021; Jeon et al., 2021; Li et al., 2022; Zhao et al., 2020; Dang et al., 2021), the adversary we consider is an honest-but-curious server with the goal of uncovering client-side labels, which gets access to the global model and shared gradients as shown in Fig 1(a). Our attack targets model architectures that contain

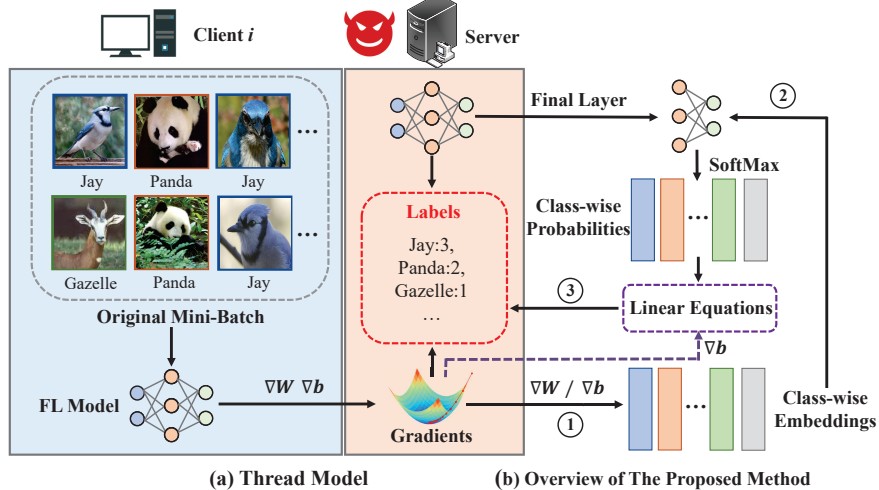

Figure 1: Illustration of the threat model and the proposed method.

at least one fully-connected layer and have a softmax activation with cross-entropy loss for classification, such as fully-connected neural networks (FCN) and convolutional neural networks from shallow to deep——LeNet-5 (LeCun et al., 1998), VGG (Simonyan & Zisserman, 2014), ResNet (He et al., 2016), etc.

## 2.2 ANALYTIC LABEL RESTORATIONS

Prior analytical label restoration methods primarily rely on a major observation. For a single sample, the derivative of the cross-entropy loss $\mathcal{L}$ w.r.t. the network final logit $\mathbf{z}$ at index $i$ is $\nabla z_i = p_i - y_i$ (See Appendix A for detailed derivation). This, obviously, leads to a unique negative sign for $\nabla z_i$ at the ground truth index $c$. However, we are able to access $\nabla \mathbf{W}_i$ instead of $\nabla z_i$. While using the chain rule, we have $\nabla \mathbf{W}_i = \nabla z_i \times \nabla_{\mathbf{W}_i} z_i = \nabla z_i \mathbf{e}^\top$. As the embedding $\mathbf{e}$ is independent of the class index $i$, the uniqueness of $\nabla z_i$'s sign is passed on to $\nabla \mathbf{W}_i$. Formally, we have $\nabla \mathbf{W}_i \cdot \nabla \mathbf{W}_j = ||\mathbf{e}||^2 \nabla z_i \nabla z_j$, $\nabla z_c < 0$, and $\nabla z_{i \neq c} > 0$, such that the label $c$ can be identified by inspecting whether $\nabla \mathbf{W}_i \cdot \nabla \mathbf{W}_j \leq 0, \forall j \neq i$. Because of the common use of non-negative activation functions (ensure that $||\mathbf{e}|| > 0$), e.g., ReLU (Glorot et al., 2011) and Sigmoid, we can simply extract the ground-truth label whose $\nabla \mathbf{W}_i$ is negative. This is exactly what iDLG (Zhao et al., 2020) does. Subsequently, the researchers (Yin et al., 2021) observe empirically the magnitude of the negative gradient significantly exceeds that of the positive gradient assuming a non-negative non-linear activation function is applied, which indicates a negative sign can still stand out after averaging operations. Therefore, they search for negative signs using minimum values rather than summation along the feature dimension to perform label restoration from batch-averaged gradients. However, without the assumption that $||\mathbf{e}|| > 0$, neither approach is viable any longer. Another batch label restoration method, Revealing Labels from Gradients (RLG) (Dang et al., 2021), offers a novel insight thus no need to satisfy that assumption. $\nabla \mathbf{W}^\top$ can be decomposed into $\mathbf{P}\Sigma\mathbf{Q}$ by singular value decomposition, where $\mathbf{P} \in \mathbb{R}^{m \times S}$ and $\mathbf{Q} \in \mathbb{R}^{S \times C}$ are orthogonal matrices, $\Sigma \in \mathbb{R}^{S \times S}$ in the middle is a diagonal matrix with non-negative elements on the diagonal, and $S = rank(\nabla \mathbf{W}^\top) < min\{m, C\}$. If we denote $\mathbf{r} = \nabla z_i \mathbf{Q}^\top$, then $\mathbf{rQ} = \nabla z_i$, which means $\mathbf{rq}^c < 0$ and $\mathbf{rq}^{j \neq c} > 0$ ($\mathbf{q}^j$ is the $j$-th column in $\mathbf{Q}$). The problem of label recovery is then transformed into finding a classifier to separate $\mathbf{q}^c$ from $\mathbf{q}^{j \neq c}$ by linear programming. Moreover, $\nabla \mathbf{W}$ can also be disassembled: $\nabla \mathbf{W} = \frac{1}{K} \sum_{j=1}^{K} \nabla \mathbf{z}^j \mathbf{e}^{\top j} = \mathbf{Z}\mathbf{E}^\top$, where $\mathbf{Z} = [\nabla \mathbf{z}^1, ..., \nabla \mathbf{z}^K] \in \mathbb{R}^{C \times K}$, $\mathbf{E} = \frac{1}{K}[\mathbf{e}^1, ..., \mathbf{e}^K] \in \mathbb{R}^{m \times K}$ and $K$ is the batch size. Assume that $\mathbf{Z}$ and $\mathbf{E}$ are full-rank matrices, we have $K = S < min\{m, C\}$, so the approach requires $K$ not to be large.

## 2.3 Single Embedding Reconstruction

In deep neural network architectures, the fully-connected layer is more vulnerable to leakage from gradients for its simple design. A recent work, i.e., InvertingGradients (IG) (Geiping et al., 2020), has brought theoretical insights on this task by showing provable embedding reconstruction feasibility.

**Theorem 1.** *For neural networks with a biased fully-connected layer (e.g., the final classification layer), presume the derivative of the loss $\mathcal{L}$ w.r.t. to the layer's output $\mathbf{z}$ contains at least one non-zero element, then the input to the fully-connected layers $\mathbf{e}$ can be uniquely reconstructed by analytic computation.*

*Proof.* Consider the mapping $\mathbf{z} = \mathbf{We} + \mathbf{b}$ of a biased full-connected layer without a nonlinear activation, it's easy to observe that $\frac{\partial \mathbf{z}}{\partial \mathbf{We}} = \frac{\partial \mathbf{z}}{\partial \mathbf{b}} = \mathbf{1} \in \mathbb{R}^C$. Due to our assumption guarantees $\frac{\partial \mathcal{L}}{\partial z_i} \neq 0$ for some index $i$, we have $\frac{\partial \mathcal{L}}{\partial b_i} = \frac{\partial \mathcal{L}}{\partial z_i} \times \frac{\partial z_i}{\partial b_i} = \frac{\partial \mathcal{L}}{\partial z_i} \times 1 = \frac{\partial \mathcal{L}}{\partial z_i}$ and $\frac{\partial \mathcal{L}}{\partial \mathbf{W}_i} = \frac{\partial \mathcal{L}}{\partial z_i} \times \frac{\partial z_i}{\partial \mathbf{W}_i} = \frac{\partial \mathcal{L}}{\partial z_i}\mathbf{e}^\top = \frac{\partial \mathcal{L}}{\partial b_i}\mathbf{e}^\top$ according to the chain rule. Therefore, $\mathbf{e}$ can be calculated exactly as $\mathbf{e} = (\frac{\partial \mathcal{L}}{\partial b_i})^{-1}(\frac{\partial \mathcal{L}}{\partial \mathbf{W}_i})^\top$. □

On the basis of Theorem 1, we are able to perfectly accomplish the analytic reconstruction of a single input to a fully-connected layer. Such a one-shot theoretical approach, however, cannot be extended directly to recover batch embeddings for unignorable information loss from the average operation.

## 3 Methodology

In this section, we propose a method to restore instance-wise labels via batch-averaged gradients, which we refer to as instance-wise Labels Restoration from Gradients (iLRG). Our method consists of three main steps as shown in Fig1 (b): (1) Reconstruct the class-wise embeddings by calculating the quotient of two gradients of the weight and bias in the final classification layer; (2) Feed the embeddings into this layer to obtain the subsequent post-softmax probabilities; (3) Solve a system of linear equations for the number of instances per class by the Moore-Penrose pseudoinverse algorithm.

### 3.1 Class-wise Embedding Reconstruction

Two crucial observations presented in Soteria (Sun et al., 2021) push the embedding reconstruction towards deeper: (1) intra-class uniformity and concentration of embedding distribution; (2) inter-class low entanglement of gradient contributions, an empirical observation that $i$-class samples mainly contribute to the $i$-th gradient row. Leveraging them, we can attempt to reconstruct the average embeddings of each class, i.e., class-wise embeddings.

We first divide a training batch $\mathbb{B}$ into subsets of $C$ distinct classes, i.e., $\mathbb{B} = \{\mathbb{B}_1, ..., \mathbb{B}_C\}$. Then, based on the observations mentioned above, we make two approximations. The first approximation has been formalized in the Appendix material of Soteria. We then further propose a theoretical account of the latter.

**Approx 1 (Intra-class Uniformity and Concentration of Embedding Distribution).** *The distribution of embeddings $\mathbf{e}$ is uniform and concentrated over a certain class of samples $\mathbb{B}_i$ in a training batch, such we can replace them with the arithmetic mean of this categoty, i.e., the geometric center.*

Consequently, the average gradient at index $i$ over $\mathbb{B}_i$ can be represented as formula 1 with Theorem 1:

$$\overline{\frac{\partial \mathcal{L}}{\partial \mathbf{W}_i}}_{\mathbb{B}_i} = \frac{1}{|\mathbb{B}_i|} \sum_{j \in \mathbb{B}_i} \frac{\partial \mathcal{L}^j}{\partial \mathbf{W}_i^j} = \frac{1}{|\mathbb{B}_i|} \sum_{j \in \mathbb{B}_i} \frac{\partial \mathcal{L}^j}{\partial b_i^j} \mathbf{e}^{j\top} \approx \left( \frac{1}{|\mathbb{B}_i|} \sum_{j \in \mathbb{B}_i} \frac{\partial \mathcal{L}^j}{\partial b_i^j} \right) \left( \frac{1}{|\mathbb{B}_i|} \sum_{j \in \mathbb{B}_i} \mathbf{e}^{j\top} \right)$$
$$= \overline{\frac{\partial \mathcal{L}}{\partial b_i}}_{\mathbb{B}_i} \overline{\mathbf{e}}_{\mathbb{B}_i}^{\top}.$$
$$(1)$$

where $\overline{(\cdot)}_{\mathbb{B}_i}$ and $(\cdot)_i^j$ denote the mean of a variable and the variable at index $i$ for sample $j$ across $\mathbb{B}_i$, respectively.

Untrained models are obviously more likely to satisfy this property. If we project the embeddings onto a 2D plane, their distribution over the entire batch is close to a uniform circle due to the rather poor classification ability. We discover that any embedding will yield a nearly uniform $1/n$ probability output at the beginning of training, where $n$ is the number of classes. When the model is well-trained, although the various categories can be well separated, the internal distribution of a certain category may not be sufficiently uniform and symmetrical, and there will be some outliers.

We already know that $\frac{\partial \mathcal{L}}{\partial b_i} = \frac{\partial \mathcal{L}}{\partial z_i} = p_i - y_i$, which means the gradient only goes negative at the ground-truth class index $c$, and $\sum_{i=1}^{C} \frac{\partial \mathcal{L}}{\partial b_i} = \sum_{i=1}^{C} (p_i - y_i) = 0$. In other words, the *av* (*absolute value*) of the negative gradient at class index $c$ is equal to the sum of *av*s of the other positive gradients. Based on this derivation, another approximation is proposed.

**Approx 2 (Inter-class Low Entanglement of Gradient Contributions).***The batch-averaged gradient row at index $i$ is mainly from $i$-class samples in a training batch. Specifically, we have:*

$$\overline{\frac{\partial \mathcal{L}}{\partial b_i}}_{\mathbb{B}} = \frac{1}{|\mathbb{B}|} \sum_{j=1}^{C} |\mathbb{B}_j| \overline{\frac{\partial \mathcal{L}}{\partial b_i}}_{\mathbb{B}_j} \approx \frac{|\mathbb{B}_i|}{|\mathbb{B}|} \overline{\frac{\partial \mathcal{L}}{\partial b_i}}_{\mathbb{B}_i}, \quad \overline{\frac{\partial \mathcal{L}}{\partial \mathbf{W}_i}}_{\mathbb{B}} = \frac{1}{|\mathbb{B}|} \sum_{j=1}^{C} |\mathbb{B}_j| \overline{\frac{\partial \mathcal{L}}{\partial \mathbf{W}_i}}_{\mathbb{B}_j} \approx \frac{|\mathbb{B}_i|}{|\mathbb{B}|} \overline{\frac{\partial \mathcal{L}}{\partial \mathbf{W}_i}}_{\mathbb{B}_i}.$$
$$(2)$$

Since $\frac{\partial \mathcal{L}}{\partial \mathbf{W}_i} = \frac{\partial \mathcal{L}}{\partial b_i} \mathbf{e}^{\top}$, the latter in formula 2 requires the variance of $||\mathbf{e}||$ over the whole batch to be smaller than the proportionality of the bias gradient. Due to the commonly used Input Normalization and Batch Normalization operations, this requirement is not difficult to meet, especially for untrained models. According to our derivation that $\nabla b_i = \nabla z_i = p_i - y_i$, the bias gradient of $i$-class sample at index $i$ is $p_i^i - 1$, while that of another class $j$ is $p_i^j$. Here the superscripts correspond to the categories. Therefore, Approx 2 holds when $|p_i^j| \ll |p_i^i - 1|$ for any $j \neq i$. Hence, the entanglement is related to the label distribution, i.e., it should not be extreme disparate. Furthermore, the magnitude of the gradients will be significantly reduced and get more sensitive to errors as training progresses.

On the basis of Approx 2, we can derive that $\overline{\frac{\partial \mathcal{L}}{\partial b_i}}_{\mathbb{B}_i}^{-1} \times \overline{\frac{\partial \mathcal{L}}{\partial \mathbf{W}_i}}_{\mathbb{B}_i}^{\top} \approx \overline{\frac{\partial \mathcal{L}}{\partial b_i}}_{\mathbb{B}}^{-1} \times \overline{\frac{\partial \mathcal{L}}{\partial \mathbf{W}_i}}_{\mathbb{B}}^{\top}$. Of course, it is necessary to ensure that $\overline{\frac{\partial \mathcal{L}}{\partial b_i}}_{\mathbb{B}} \neq 0$ and $\overline{\frac{\partial \mathcal{L}}{\partial b_i}}_{\mathbb{B}_i} \neq 0$ here. The occurrence of zero is uncommon, but when it does occur, we replace it with a small enough number $\epsilon$. Combined with formula 1, we finally get $\overline{\mathbf{e}}_{\mathbb{B}_i} \approx \overline{\frac{\partial \mathcal{L}}{\partial b_i}}_{\mathbb{B}}^{-1} \times \overline{\frac{\partial \mathcal{L}}{\partial \mathbf{W}_i}}_{\mathbb{B}}^{\top}$ to reconstruct the class-wise embeddings.

**Bypass Approx 2 when untrained.** In Approx 2, if the error terms for weight and bias gradients are not ignored, we have:

$$\overline{\frac{\partial \mathcal{L}}{\partial b_i}}_{\mathbb{B}} = \frac{1}{|\mathbb{B}|} \left( |\mathbb{B}_i| \overline{\frac{\partial \mathcal{L}}{\partial b_i}}_{\mathbb{B}_i} + \sum_{j \neq i} |\mathbb{B}_j| \overline{\frac{\partial \mathcal{L}}{\partial b_i}}_{\mathbb{B}_j} \right), \quad \overline{\frac{\partial \mathcal{L}}{\partial \mathbf{W}_i}}_{\mathbb{B}} = \frac{1}{|\mathbb{B}|} \left( |\mathbb{B}_i| \overline{\frac{\partial \mathcal{L}}{\partial \mathbf{W}_i}}_{\mathbb{B}_i} + \sum_{j \neq i} |\mathbb{B}_j| \overline{\frac{\partial \mathcal{L}}{\partial \mathbf{W}_i}}_{\mathbb{B}_j} \right).$$
$$(3)$$

For an untrained model, the average $\mathbf{e}$ over any $\mathbb{B}_i$ is almost equal. As a result, we have:

$$\overline{\frac{\partial \mathcal{L}}{\partial \mathbf{W}_i}}_{\mathbb{B}} \approx \frac{1}{|\mathbb{B}|}(|\mathbb{B}_i|\overline{\frac{\partial \mathcal{L}}{\partial b_i}}_{\mathbb{B}_i} + \sum_{j \neq i}|\mathbb{B}_j|\overline{\frac{\partial \mathcal{L}}{\partial b_i}}_{\mathbb{B}_j})\mathbf{e}^\top. \tag{4}$$

Since the restored embedding is the quotient of the two formula 4 and the former of 3, the error terms hardly affect the proportional result in this case, which means we can bypass Approx 2. Taking these two approximations together, we assert that our attacks on untrained models outperform trained models.

The improvement of our work over Soteria is that they just restore a linearly scale embedding $\gamma \nabla \mathbf{W}_i$ while ours is more detailed and precise, where $\gamma$ is a scale influences by the local training steps ($\gamma = 1$ in FedSGD and $\gamma > 1$ in FedAVG (McMahan et al., 2016)) and $\nabla \mathbf{W}_i$ is a brief notation for $\overline{\frac{\partial \mathcal{L}}{\partial \mathbf{W}_i}}_{\mathbb{B}}$. The reason is that they don't apply Theorem 1 to make use of $\nabla b_i$.

## 3.2 INSTANCE-WISE LABEL RESTORATION

If we extend $\nabla \mathbf{z}_i = p_i - y_i$ for single sample to the whole batch, we can get $\sum_k \frac{\partial \mathcal{L}^k}{\partial z_i^k} = \sum_k p_i^k - \sum_k y_i^k$, where $(\cdot)_i^k$ denotes the variable at index $i$ for sample $k$ in a batch of size $K$. After adjusting the order, it becomes $\sum_k p_i^k - \sum_k \frac{\partial \mathcal{L}^k}{\partial z_i^k} = \sum_k y_i^k$. The right of this equation is exactly the total number of instances of class $i$ in a batch. Let $k_j$ denotes the number of $j$-class instances, and we can disassemble $\sum_k p_i^k$ into $\sum_j k_j \overline{p}_{i\mathbb{B}_j}$. From the class-wise embeddings that was previously deduced, it is possible to recover $\overline{\mathbf{p}}_{\mathbb{B}_j}$, again based on the intra-class uniformity and concentration of embedding distribution.

**Approx 3 (Average Probabilities from Average Embeddings).** *The average post-softmax probability from j-class samples by classification model with softmax activation is approximate to that produced by j-class average embedding.*

$$\overline{\mathbf{p}}_{\mathbb{B}_j} = \frac{1}{|\mathbb{B}_j|}\sum_{t \in \mathbb{B}_j} SoftMax(\mathbf{We} + \mathbf{b}) \approx SoftMax(\mathbf{W}\overline{\mathbf{e}}_{\mathbb{B}_j} + \mathbf{b}). \tag{5}$$

Additionally, we have $\sum_k \frac{\partial \mathcal{L}^k}{\partial z_i^k} = K\overline{\frac{\partial \mathcal{L}}{\partial z_i}}_{\mathbb{B}} = K\overline{\frac{\partial \mathcal{L}}{\partial b_i}}_{\mathbb{B}} = K\nabla b_i$, where $K$ is the batch size. Therefore, we arrive to an equation:

$$\sum_j k_j \overline{p}_{i\mathbb{B}_j} - K\nabla b_i = k_i, i = 1, ..., C, j = 1, ..., C. \tag{6}$$

This is equivalent to a system of equations as shown in (7):

$$\begin{cases} k_1 + ... + k_i + ... + k_C = K, \\ \sum_j k_j \overline{p}_{1\mathbb{B}_j} - K\nabla b_1 = k_1, \\ \sum_j k_j \overline{p}_{2\mathbb{B}_j} - K\nabla b_2 = k_2, \\ \qquad\qquad \vdots \\ \sum_j k_j \overline{p}_{C\mathbb{B}_j} - K\nabla b_C = k_C. \end{cases} \tag{7}$$

Since the coefficient matrix of this system is not square, we adopt the Moore-Penrose pseudo-inverse algorithm to get a LSS. And the final result also requires filtering of outliers and rounding. If the occurences of the class-wise labels can be obtained in advance, we can further simplify the above equation system through the prior works (Zhao et al., 2020; Yin et al., 2021; Dang et al., 2021), i.e., to discard unknowns $k_j$ and the corresponding equation where there is no $j$-class sample in the training batch.

## 4 EXPERIMENTS

**Setups.** We evaluate our method for the classification task on three classic image datasets with ascending classes and four models from shallow to deep: (1) a 3-layer FCN (Fully-Connected Network) and other variations on the MNIST dataset with 10-class grayscale images of size 28×28. (2) the 7-layer LeNet-5 on the CIFAR100 dataset with 100-classes color images of size 32×32. (3) the 16-layer VGG-16 and other variations on the large-scale 1000-class ImageNet ILSVRC 2012 dataset (Deng et al., 2009) at 224 ×224 pixels. (4) the 50-layer ResNet-50 and other variations on the ImageNet dataset. We use the training set by default in the following experiments, and the images have been normalized during data loading stage. All statistics except those in Section 4.1 are averaged in a randomly selected batch on 50 repetitive tests. As mentioned previously, our attack is more effective on untrained models. Therefore, unless otherwise specified, we focus on the untrained model.

**Evaluation metrics.** To quantitatively analyze the performance of our label restoration attack, we propose the following two metrics: (1) *Label existence Accuracy—LeAcc*: the accuracy score for predicting label existences; (2) *Label number Accuracy—LnAcc*: the accuracy score for predicting the number of instances per class. Furthermore, in Section 4.4, we adopt the image reconstruction quality score *Peak Signal to Noise Ratio* (*PSNR*) and the perceptual similarity score *Learned Perceptual Image Patch Similarity* (*LPIPS*) (Zhang et al., 2018) to measure the similarity between the restored images and the ground truth.

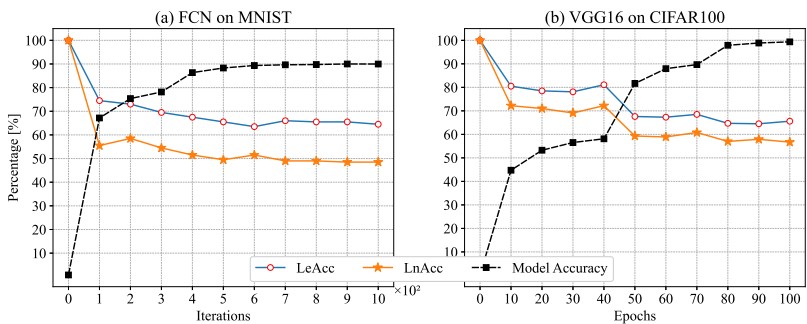

Figure 2: Performance comparison at different training stages.

### 4.1 COMPARISON OF DIFFERENT TRAINING STAGES

We first compare the attack results at different training stages as shown in Fig 2. We perform attacks on batches of size 64 and 8 on MNIST and CIFAR 100 with the FCN-3 and VGG-16 models, respectively. All LeAccs and LnAccs are the average of 20 repetitive experiments. Consistent with the analysis in Section 3.1, the attack effect gets worse as training progresses. Moreover, The corresponding error analysis is given in Appendix D. Approx 1 is generally satisfied for both untrained and well-trained models, thus we speculate that the drop may be due to the larger error contributed by Approx 2.

### 4.2 COMPARISON WITH PRIOR WORKS

**Attack Baselines.** We compare our attack with prior analytic approaches: (1) Improved Deep Leakage from Gradients (iDLG) (Zhao et al., 2020): the single-sample label inference by finding the gradient row whose dot product with other columns yields a negative value; (2) GradInversion (GI) (Yin et al., 2021): the label restoration for the case of single-instance per class by the order of the minimum element in each gradient row; (3) Revealing Labels from Gradients(RLG) (Dang et al., 2021): the extraction of a class-wise label set based on Singular Value Decomposition (SVD) and Linear Programming (LP). Owing to the limitations that iDLG and GI share, we scale them to the real-world settings of multiple instances per class in a batch for comparisons. For iDLG, we alter the selection of the gradient row with the smallest element after summation to all negative gradient rows. And we pick all negative gradients instead of the Top-$K$ minimums in GI, where $K$ is the batch size.

Table 1: Quantitative Comparison of our label restoration attack with prior works on diverse scenarios. The batch size remains at 24. *: replace the activation function in LeNet-5 from ReLU to Swish (Ramachandran et al., 2017) which is applied in EfficientNet (Tan & Le, 2019); - : this metric is incapable of being calculated. CosSim: the Cosine Similarity for our recovered post-softmax probabilities.

| | | iDLG | | GI | | RLG | | **Ours** | | |
|---|---|---|---|---|---|---|---|---|---|---|
| **Model** | **Dataset** | LeAcc | LnAcc | LeAcc | LnAcc | LeAcc | LnAcc | LeAcc | LnAcc | CosSim |
| FCN-3 | MNIST | 0.514 | - | 1.000 | - | 0.932 | - | **1.000** | **0.994** | **0.979** |
| LeNet-5 | CIFAR100 | 1.000 | - | 1.000 | - | 1.000 | - | **1.000** | **1.000** | **1.000** |
| LeNet-S* | CIFAR100 | 0.150 | - | 0.213 | - | 1.000 | - | **1.000** | **1.000** | **1.000** |
| VGG-16 | ImageNet | 1.000 | - | 1.000 | - | 0.981 | - | **1.000** | **1.000** | **1.000** |
| ResNet-50 | ImageNet | 1.000 | - | 1.000 | - | 1.000 | - | **1.000** | **1.000** | **1.000** |

**Validity of Probability Reconstruction.** The key to our approach is the estimation of the class-wise average probabilities. Our results of the last column in Table 1 show that recovery of the class-wise probability is quite precise and corroborates with the label restoration accuracies. We even achieve a 100% CosSim on both CIFAR100 and ImageNet datasets, and the essential cause behind the fact is that almost each class consists of at most two or three samples in a random-selected batch of size 24.

**Results of Label Restoration.** Table 1 compares the performance of our proposed iLRG with several prior attack methods. First and foremost, we drill down the ability of the attack from determining the presence of labels to the number of instances for each class while the others are incapable of that. Our method achieves over 99.0% on the whole models and datasets for the two evaluation metrics here. In terms of LeAcc, GI performs best except our iLRG when a non-negative nonlinear activation function is utilized in the model. Nevertheless, once the activation in the network contains a negative element, its performance and that of iDLG will plummet. The essential cause of this fact has been expounded in Section 2.2.

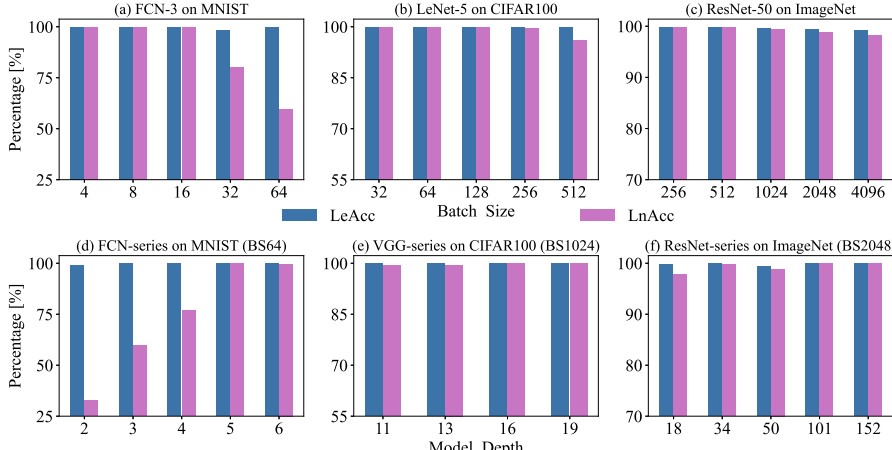

Figure 3: The percentages of LeAcc and LnAcc as batch size and model depth increase under various settings. The FCN series models are distinguished by the number of hidden layers, where a 2-layer model has no hidden layers and the basic FCN-3 consists of a hidden layer (See Appendix B for the FCN-3 network architecture). And BS is short for batchsize.

## 4.3 EFFECTS OF BATCH SIZE AND MODEL DEPTH

We next discuss the effect of increasing Batch Size and Model Depth/Layer Amount on the results. From Fig 3, we can observe that: (1) the LeAcc score remains above 99% under all experimental settings; (2) as we raise the batch size, the attack capability decreases due to the greater information

loss caused by average operation; (3) the depth or complexity of a model likewise plays a strong impact factor. The attack performance increases as the network deepens, which can be seen from the fact that the ResNet-series models are more vulnerable to our attack on large batches than the shallow FCN-series. The results of various networks within the same family also confirm the claim. In Fig 3 (f), it appears to contradict this that the attack against ResNet-50 is inferior to that of ResNet-34. However, since the basic component of Resnet-50 and deeper networks switches from BasicBlock to BottleNeck, the number of parameters of ResNet-50 is actually less than ResNet-34.

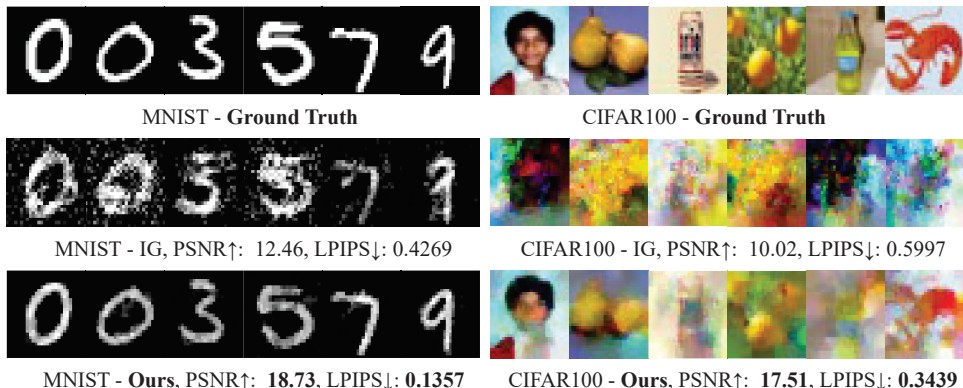

Figure 4: Batch image reconstruction on MNIST (FCN-3, BS50) and CIFAR100 (ResNet-18, BS16) compared with IG (Geiping et al., 2020). We assign a specific label to each instance after label restoration at 100% accuracy. The 6 best visual images are selected to display and calculate the metrics.

## 4.4 IMPROVED GRADIENT INVERSION ATTACK WITH OURS

Gradient inversion attacks perform joint optimization of model inputs and labels, thus labels may shift during optimization, which commonly leads to poor recovery of the inputs. Naturally, it occurs to us that the proposed method can be used to specify an optimization objective for each instance, so as to enhance the existing attacks. Since the adversary randomly initializes the dummy inputs of a batch as the optimization objectives in the gradient inversion attack, which is originally unordered, we can assign any label to each instance according to the instance-wise label restoration results and they will eventually produce exactly the same batch-averaged gradients. Therefore, for each category with at least one instance, we simply select a subset with the capacity of the number of instances of this class from the remaining randomly initialized dummy inputs and assign them labels of this category. Fig 4 illustrates the improvement both visually and numerically. We choose IG as the baseline because it does not require substantial prior constraints.

## 5 CONCLUSIONS

This work proposes *instance-wise Label Restoration from Gradients* (*iLRG*), a method to reveal instance-wise labels via shared batch-averaged gradients in FL. We build and solve a system of linear equations over the labels by leveraging a crucial derivation about the gradient of the network final logits and an approximate reconstruction of class-wise averaged probabilities. Our method performs extremely well for untrained models. We conduct comprehensive experiments on three classic image datasets with ascending classes and four models from shallow to deep (e.g., FCN on MNIST, LeNet-5 on CIFAR100, VGG and ResNet on ImageNet, etc) under the various settings of a large batch size up to 4096. The evaluations demonstrate the capability of iLRG with a high proportion of both *Label existence Accuracy* (*LeAcc*) and *Label number Accuracy* (*LnAcc*). It works even on models with an activation function that is not uniformly non-negative. Finally, We further facilitate the existing gradient inversion attacks by exploiting the recovered labels.

ACKNOWLEDGMENTS

This work was supported by the National Natural Science Foundation of China (32071775, U21B2021). Finally, Kailang Ma would like to extend great gratitude to my younger colleague Gaojian Xiong for his contribution to the theoretical aspect of this paper.

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

## A    THE DERIVATION OF $\nabla z_i = \dfrac{\partial \mathcal{L}}{\partial z_i} = p_i - y_i$

According to the definitions and notations given in Section 2.1, we have $\mathbf{z} = \mathbf{W}\mathbf{e} + \mathbf{b}$ and $\mathbf{p} = SoftMax(\mathbf{z})$ to model the mapping of the final layer and post-softmax process. First, the softmax function defines a transformation that $p_i = SoftMax(\mathbf{z})_i = \dfrac{e^{z_i}}{\sum_{j=1}^{C} e^{z_j}}, j = 1, ..., C$. We denote $\sum_{j=1}^{C} e^{z_j}$ as $\sum_C$ for convenience and discuss $\dfrac{\partial p_i}{\partial z_j}$ by situation:

$$\frac{\partial p_i}{\partial z_j} = \begin{cases} \dfrac{e^{z_i}\sum_C - e^{z_i} e^{z_i}}{\sum_C^2} = \dfrac{e^{z_i}}{\sum_C} - \dfrac{e^{z_i}{}^2}{\sum_C} = p_i - p_i{}^2, & i = j, \\[2ex] \dfrac{0 \times \sum_C - e^{z_i} e^{z_j}}{\sum_C^2} = -\dfrac{e^{z_i}}{\sum_C} - \dfrac{e^{z_j}}{\sum_C} = -p_i p_j, & i \neq j, \end{cases} \tag{8}$$

where $i$ and $j$ are both members of $\mathbb{C} = \{1, ..., C\}$. And the cross-entropy loss can be formalized as $\mathcal{L} = CE(\mathbf{p}, \mathbf{y}) = -\sum_{i=1}^{C} y_i log(p_i) = -log(p_c)$ ($c$ represents the index of the ground-truth class), which means $\dfrac{\partial \mathcal{L}}{\partial p_i} = \dfrac{\partial \mathcal{L}}{\partial p_c} = -\dfrac{1}{p_c}$ only if $i = c$ otherwise 0. Therefore, using the chain rule, $\dfrac{\partial \mathcal{L}}{\partial z_i}$ can be calculated as the following formula:

$$\frac{\partial \mathcal{L}}{\partial z_i} = 0 + \frac{\partial \mathcal{L}}{\partial p_c} \times \frac{\partial p_c}{\partial z_i} = \begin{cases} -\dfrac{1}{p_c} \times (p_c - p_c{}^2) = p_c - 1, & i = c, \\[2ex] -\dfrac{1}{p_c} \times (-p_c \times p_i) = p_i, & i \neq c. \end{cases} \tag{9}$$

Merging the two branches, we get our conclusion that $\nabla z_i = \dfrac{\partial \mathcal{L}}{\partial z_i} = p_i - y_i$.

## B    THE ARCHITECTURE OF FCN-3

See Table 2.

Table 2: The details of shape for each FCN-3 layer.

| Layer | Shape (In-Out) |
|-------|----------------|
| Input | 784-300 |
| Hidden | 300-300 |
| Output | 300-10 |

## C    PSEUDO-CODE OF OUR ALGORITHM

Algorithm 1 provides a pseudo-code for the complete procedure of our method.

## D    APPROXIMATION ERROR ANALYSIS

For the experiment in Section 4.1, we conduct error analysis shown in the Table 3 below. We choose the models at epoch 0, 40 and 100 respectively as the representatives of the three stages of untrained, mid-training and well-trained. The three approximations and restored embeddings are our check items. And the evaluation metrics include MSE (Mean Squared Error), MRE (Mean Relative Error) and CosSim (Cosine Similarity), where MRE is the ratio of the error to the ground-truth.

---

**Algorithm 1** Class-Wise Embeddings Inference and Instance-Wise Batch Label Restoration.

---

**Input:** Gradients of weight and bias in the final layer $\nabla \mathbf{W} \in \mathbb{R}^{C \times m}, \nabla \mathbf{b} \in \mathbb{R}^{C}$.
**Output:** Class-wise averaged embeddings $\mathbb{E} = \{\mathbf{e}^{i}, i = 1, 2, .., C\}$ and the number of occurences
     for each class $\mathbb{N} = \{\mathbf{n}^{i}, i = 1, 2, .., C\}$ in a training batch.
 1: Initial $\mathbb{E} = \emptyset$ and $\mathbb{N} = \emptyset$.
 2: **for** $i = 1$ to $C$ **do**
 3:      Calculate $\mathbf{e}^{i}$ using $\nabla \mathbf{W}$ and $\nabla \mathbf{b}$;
 4:      Feed $\mathbf{e}^{i}$ into the final layer to get the network outputs $\mathbf{z}^{i}$ and post-softmax probabilities $\mathbf{p}^{i}$;
 5:      Add $\mathbf{e}^{i}$ into $\mathbb{E}$;
 6: **end for**
 7: Solve the system of linear equations in (7) to obtain $\mathbb{N}$.
 8: **return** $\mathbb{E}$ and $\mathbb{N}$.

---

Table 3: Comparison of the errors at different training stages.

| Training Stage | Approx 1 | | Approx 2 | | Approx 3 | | Embedding | |
| --- | --- | --- | --- | --- | --- | --- | --- | --- |
| | MSE | MRE | MSE | MRE | MSE | MRE | MSE | CosSim |
| Untrained (Epoch 0) | **1.6e-11** | **0.000** | 9.6e-5 | **1.377** | **6.1e-16** | **0.000** | **1.6e-5** | **0.978** |
| | | | 2.4e-8 | 2.896 | | | | |
| Middle (Epoch 40) | 6.6e-5 | 0.668 | 5.7e-5 | 3.615 | 9.1e-5 | 0.157 | 1.8e-1 | 0.739 |
| | | | 6.5e-7 | **1.391** | | | | |
| Trained (Epoch 100) | 2.8e-6 | 0.822 | **3.0e-7** | 13.218 | 2.6e-6 | 0.130 | 5.3e-1 | 0.734 |
| | | | 4.0e-8 | 6.129 | | | | |

First of all, both Approx1 and Approx3 are related to *Intra-class Uniformity and Concentration of Embedding Distribution*. Their MSE results are consistent with this property: best at the beginning, second at the end, and worst in the middle. However, since the gradient magnitude becomes smaller as the training progresses, the mid-term MRE may be less than the final (for Approx1 here).

Secondly, Approx2 represents *Inter-class Low Entanglement of Gradient Contributions*. Due to the complexity of weight gradient errors, we choose to analyze the bias gradient. As the training progresses, MSE decreases but MRE increases, indicating that the entanglement is actually increasing.

Finally, we note that the recovered embeddings for untrained model are the best. This is because it satisfies both properties and can even bypass Approx 2. In addition, we can also assert from the above results that the error of Approx 2 may be larger than that of Approx1. Since Approx 2 directly participates in the division calculation for restoring embeddings, which intuitively may also be a reason.

Table 4: Comparison of the embedding reconstruction quality.

| Training Stage | Soteria | | Ours | |
| --- | --- | --- | --- | --- |
| | MSE | CosSim | MSE | CosSim |
| Untrained (Epoch 0) | 2.6e-4 | **0.978(-)** | **1.6e-5** | 0.978 |
| Middle (Epoch 40) | **2.0e-2** | 0.599(-) | 1.8e-1 | **0.739** |
| Trained (Epoch 100) | **1.9e-1** | 0.396(-) | 5.3e-1 | **0.734** |

# E   COMPARISON WITH SOTERIA

We compare the quality of class-wise embeddings at different training stages by Soterta (Sun et al., 2021) and the proposed method's. The experimental setup remains the same as in Section 4.1 and Appendix D. Here (-) indicates that it is originally a negative value. As shown in the Table 4, our

method outperforms Soteria in terms of the cosine similarity metric. This suffices to demonstrate that our restorations are of high quality. Moreover, we note that our restored embeddings are more precise than Soteria's when untrained, but it doesn't hold up well after training. In fact, MSE of the proposed method is almost always less than that of Soteria. Only in rare cases, a large MSE leads to a large average MSE.

Table 5: The attack effect under the extreme distribution.

| Training Stage | BatchSize | Per-Class NOI | | |
| --- | --- | --- | --- | --- |
| | | Class 0 | Class 18 | Class 92 |
| Untrained (Epoch 0) | 24 | **1** | 22 | **1** |
| | 72 | **1** | 70 | **1** |
| | 216 | **1** | 214 | **1** |
| | 648 | **1** | 646 | **1** |
| Trained (Epoch 100) | 24 | **1** | 18 | **1** |
| | 72 | **1** | 57 | **1** |
| | 216 | 10 | 155 | 9 |
| | 648 | 51 | 371 | 65 |

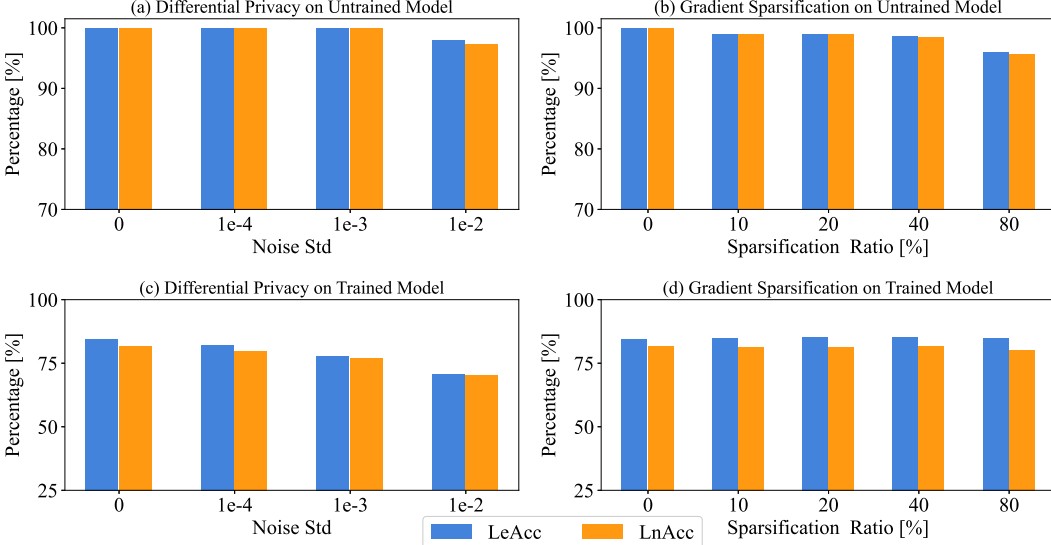

Figure 5: The impact of two typical defense strategies Differential Privacy and Gradient Sparsification on the label restoration results.

## F EFFECT OF LABEL DISTRIBUTION

The inter-class entanglement of gradient contributions is significantly influenced by the label distribution. In an extremely imbalanced case, the gradients of a minor category $i$ will be entangled with categories that consist of significantly more instances than it, i.e., the dominance of $\mathbb{B}_i$ over the gradients at index $i$ is weakened. To prove the statement, we execute attacks on CIFAR100 with both untrained and trained VGG-16 models in batches ranging from 24 to 648. The results in the Table 5 are the average recovered *NoI*(*Number of Instances*) of 20 repetitive experiments. We simply select 3 classes with class-id 0,18 and 92, whose NoIs are 1, BS-2 and 1 respectively (BS denotes the batch size). It can be seen that when the degree of imbalance exceeds a certain threshold for the trained model, the recovery effect for minor classes 0 and 92 deteriorates significantly. However, our method enables restoring labels of minor classes perfectly even for such a data distribution as

626:1 when untrained. This is actually because in this case, the errors arising from entanglement can be bypassed.

## G  EFFECT OF DEFENSE STRATEGIES

The key to defending against our attack is to avoid exchanging precise gradients. We discuss two defense schemes: (1) Differential Privacy (Additive Noise): add a Gaussian noise $\epsilon \sim \mathcal{N}(0, \sigma^2 \mathbf{I})$ to the gradients with different $\sigma$; (2) Gradient Spasification: prune the gradients based on magnitude. We execute attacks on CIFAR100 with the untrained and well-trained ResNet-18 models, the batch-size is 24 and all results are the average of 20 repetitive experiments. As shown in Fig 5, both LeAcc and LnAcc are reduced as the noise increases. For the untrained model, neither of these strategies affects the results much. Differential Privacy has a greater impact on the well-trained model, while moderate Gradient Spasification even improves its performance a little. We speculate that pruned gradients with smaller magnitudes are often insignificant and may reduce the errors yielded by using them to calculate for restoring labels.

## H  MORE VISUALIZATION EXAMPLES

As shown in Fig 6, we offer additional visual examples to illustrate our improvements of gradient inversion attacks.

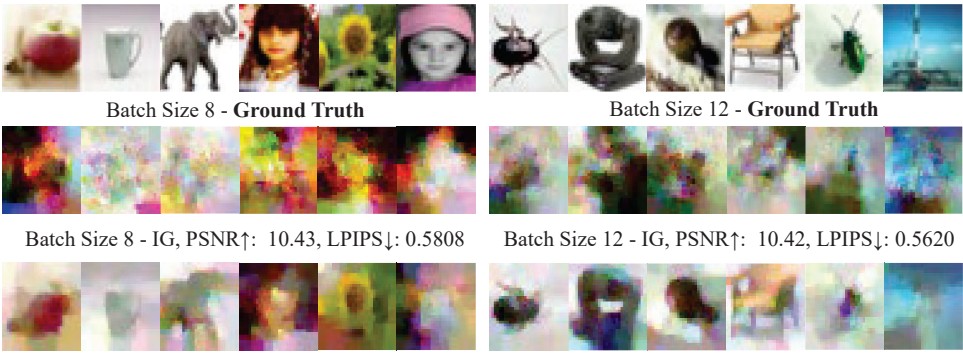

Figure 6: Additional contrast examples of inverting ResNet-18 gradients on CIFAR100 to demonstrate our improvements.

