# OpenReview forum: "Instance-wise Batch Label Restoration via Gradients in Federated Learning"
_ICLR.cc/2023/Conference — ICLR 2023 poster_

### Official Review · Reviewer_9FpD · 2022-10-16

**Confidence:** 4
**Correctness:** 2
**Technical Novelty And Significance:** 2
**Empirical Novelty And Significance:** 2
**Recommendation:** 6

**Clarity, Quality, Novelty And Reproducibility:**

Clarity: The paper is somewhat clear, but some important details are missing or unclear.

Quality: The paper appears to be technically sound.

Novelty: The paper contributes some new ideas or represents incremental advances.

Reproducibility: Key details are sufficiently well-described for competent researchers to confidently reproduce the main results.

**Strength And Weaknesses:**

Strength：
This paper considers the more real-world settings in gradient inversion attacking of Federated Learning (FL), where there are multiple instances of each class in a training batch. Besides, this paper derives the equation to design the algorithm.

Weakness：
- The paper is somewhat clear, but some important details are missing or unclear. For example, does the model of server share the same network architecture with the model of clients in Figure 1？Moreover，in Figure 2，how do servers and clients interact？Is the iLRG method run on the server side?
- The authors leverage Intra-class High Similarity and Inter-class Low Entanglement. However, this may assume that the server can obtain the labels of clients, which violates the privacy setting of FL.
- There are missing ablation experiments with the experimental part. For example, which of the intra-class high similarity and inter-class low entanglement contributes more to the result？

Question:
What judgment does the server use to determine similar or dissimilar classes in section 3.1？


**Summary Of The Paper:**

This paper presents to perform instance-wise batch label restoration from only the gradient of the final layer, to extract labels from gradients. The core idea is to establish linear equations of the gradients, probabilities and labels to derive the Number of Instances (NoI) per class by the Moore-Penrose pseudoin-verse algorithm. For this purpose, this paper designs two metrics and performs empirical studies that achieve state-of-the-art performance on three benchmark datasets.

**Summary Of The Review:**

Please see Q1 and Q2.

---

> ### Author Response · Authors · 2022-11-12
> **Response to reviewer 9FpD**
>
> Thank you for pointing out that the description of some details in this article is not clear enough. We sincerely accept your opinions and will make revisions and improvements.
>
> > **Q1: Does the model of server share the same network architecture with the model of clients in Figure 1？Moreover，in Figure 2，how do servers and clients interact？Is the iLRG method run on the server side?**
>
> Thanks for your comments. We will follow your suggestions to refine the descriptions. We have introduced the threat model in Section 2.1: **"The adversary we consider is an honest-but-curious server that has access to the global model and shared gradients, as shown in Fig 1"**. This means that we adopt a white-box attack setup (FL with a centralized server) as DLG[4], where **the adversary, i.e., the server, shares a same global model with all clients, receives the gradients from each client independently and aggregates them**. Therefore, **we only show a shared FC layer in Figure 2, where a certain client provides $\nabla \boldsymbol W$ & $\nabla \boldsymbol b$ and the server feeds the recovered embeddings into this layer to obtain the probabilities (the server performs the iLRG method)**.
>
> [4] Ligeng Zhu, Zhijian Liu, and Song Han. Deep leakage from gradients. Advances in neural information processing systems, 32, 2019.
>
> > **Q2: The method leverages Intra-class High Similarity and Inter-class Low Entanglement. However, this may assume that the server can obtain the labels of clients, which violates the privacy setting of FL.**
>
> Sorry for not understanding the meaning of this point, we're hoping you can explain it further. Intra-class High Similarity and Inter-class Low Entanglement (modified to Intra-class Uniformity and Concentration of Embedding Distribution and Inter-class Low Entanglement of Gradient Contributions) are **the two empirical properties on which we rely for embedding reconstruction (which also underlie the theoretical derivation of our approach)**, and no need to assume that the server obtains the labels of clients; instead, that is the objective of our attack. **Maybe you mean the server shouldn't know the label id-label name mapping? However, we do not think that the attack will be affected even if the mapping is not known.** For instance, we do not need to know the mapping when using the recovered labels to guide input recovery.

---

> > ### Author Response · Authors · 2022-11-12
> > **Response to reviewer 9FpD (Part 2)**
> >
> > > **Q3: There are missing ablation experiments with the experimental part. For example, which of the intra-class high similarity and inter-class low entanglement contributes more to the result?**
> >
> > Thanks for your suggestion.  However, it is difficult to perform ablation experiments, because **these two properties are only the theoretical basis of our method rather than specific components, and they are related to the training stage of the model and the data distribution in the batch, etc**. From the theoretical analysis [refer to Q3(2) & Q3(3)] and results [refer to Error Analysis] in the responses to reviewer ngm7, **the degree of entanglement has a greater impact on the results, because they directly participate in the division calculation of restoring embeddings**.
> >
> > > **Q4: What judgment does the server use to determine similar or dissimilar classes in section 3.1?**
> >
> > Thanks for your question. **The server does not have to determine whether two categories are similar or dissimilar**; it just performs a division operation before the subsequent steps to restore the class-wise average embeddings. **Furthermore, since the server is aware of the current training stage, it has a rough guess on the degree of similarity/(uniformity and concentration) within the same class, the entanglement of different classes, and the final attack effects** (the data distribution within the batch also has an impact on them, which the server cannot predict).

---

> ### Author Response · Authors · 2022-11-22
> **Looking forward to your feedback**
>
> Dear reviewer 9FpD:
>
> Thanks very much for your dedication in reviewing our paper. Your comments about some unclear details have been taken into account and prompted us to improve them in a revision of manuscript. We would appreciate it if you could re-evaluate our paper and post further valuable comments.  And sincerely hope to continue the discussion with you and address the concerns.
>
> Best regards, Authors of #1628

---

> ### Author Response · Authors · 2022-11-29
> **Follow up with reviewer 9FpD**
>
> Dear reviewer 9FpD:
>
> As the deadline for revision and discussion is approaching, we would like to follow up with your feedback. We understand you may have a busy schedule, but we believe that we have addressed all your concerns. If you still have further concerns or feel unclear after reading our responses, please kindly let us know and we are willing to make clarification and discussion. If you are satisfied with our responses so far, we sincerely hope you could consider your score. Thanks very much!
>
> Best regards, Authors of #1628

---

> ### Author Response · Authors · 2022-12-05
> **Looking forward to a discussion before the deadline**
>
> Dear reviewer 9FpD:
>
> Thanks again for your great effort in reviewing our paper! Since there are only less than **7 days** to the deadline for the second phase of discussions, we are really looking forward to having a discussion with you about all technical details and concerns. We sincerely hope to get your further feedback. Would you mind checking our response and letting us know if you have further questions?
>
> Best regards, Authors of #1628

---

> ### Author Response · Authors · 2022-12-09
> **A kindly Reminder Message: 3 days left in final discussion phase**
>
> Dear reviewer 9FpD:
>
> Hope everything is fine with you! This is a gentle reminder that there are 3 days remaining in the final discussion phase. We sincerely look forward to your response, and we are more than happy to continue the discussion for the remainder of the discussion time.
>
> Best regards, Authors of #1628

---

> ### Comment · Reviewer_9FpD · 2022-12-12
> **Acknowledgement of Rebuttal.**
>
> Thanks for the rebuttal. The authors have provided a detailed and useful rebuttal, which effectively addresses most of my concerns. As a result, I am raising my score for the paper.

---

> > ### Author Response · Authors · 2022-12-12
> > **Thanks sincerely for raising the score**
> >
> > Dear reviewer 9FpD:
> >
> > Very glad to address most of your concerns, and sincerely thanks for your raising the score! Your suggestions have significantly helped us to further improve the work.
> >
> > Best regards, Authors of #1628

---

### Official Review · Reviewer_CT8P · 2022-10-24

**Confidence:** 4
**Correctness:** 4
**Technical Novelty And Significance:** 3
**Empirical Novelty And Significance:** 3
**Recommendation:** 8

**Clarity, Quality, Novelty And Reproducibility:**

$\textbf{Clarity}$: While this paper is overall well-written, authors should try to improve the clarity of the paper, some long sentences in the manuscript make the ideas behind really hard to follow. Authors are encouraged to use intuitive and simple explanations in sections that are important for the audience such as Section 2.2 and Section 3.2.

$\textbf{Quality}$: I believe this paper is of high quality, authors provide an appendix to supplement additional information, and they released their well-structured implementation code as well as a comprehensive readme and experiment log.

$\textbf{Originality}$: A large part of this work is based on the discovery made by Sun $\textit{et. al}$ [1], however, the authors also made a non-trivial supplement to the existing work that leads to significant empirical results. Thus I believe this paper has considerable novelties.

$\textbf{Reproducibility}$: After examining the code and experiment log, this work is believed to be reproducible.




**Strength And Weaknesses:**

Pros:
1. Authors studied a non-trivial research question, which is how to recover the training labels in federated learning through the gradient inversion attack when the batch size is large and the number of classes is more than two. With such information, the features of training samples could be better inferred, hence posing more server threats to the security of federated learning.
2. Compared with existing work, this work does not need the strong assumption that the categories in the same batch must be small or equal to two.
3. This work empirically shows that the training labels can be accurately inferred even when the training batch is 4096, to the reviewer’s best knowledge, this is a new record and can without doubt boost the performances of most of the gradient inversion attacks.
4. Authors insightfully leverage the averaged post-softmax probabilities, to link the label recovery and class-wise averaged embeddings via Equation (4) and (5).

Cons:
1. Authors pointed out that existing work might fail when the activation function can yield negative values. However, I did not see how the authors' proposed method can solve this.
2. I’m interested in the situation when there is no bias for the FC layer since the absence of bias is rare - but not impossible. In that case, is the reconstructed class-wise embedding reduced to the embedding produced by [1]? Also, while the authors make the justification that leveraging bias gradients can be helpful, I recommend authors make ablation studies to compare the quality of the class-wise embedding learned by [1] and authors’ method, which could make the authors’ contribution more convincing.
3. After we know the number of instances per class within a batch, how do we assign the labels to each individual instance? If we don't do so, can authors elaborate on how only knowing the count of each class within a batch can help recover features?
4. When there are some basic defensive strategies applied (such as DP noise and gradient pruning), what are their influence on the authors' method?

The aforementioned "Cons" are not necessarily the limitations, but some clarifications/ablation studies that I'm interested in.

[1] Sun, Jingwei, et al. "Soteria: Provable defense against privacy leakage in federated learning from representation perspective." Proceedings of the IEEE/CVF conference on computer vision and pattern recognition. 2021.

**Summary Of The Paper:**

This manuscript studies the instance-level label restoration in mini-batch training under the federated learning structure. The authors proposed a strong method that is capable of restoring labels via gradient inversion attack even when batch size is as large as 4096, which was usually considered a challenging task. This is achieved by estimating the class-wise averaged embedding (to which authors improved over [1]) to estimate the class-wise averaged predicted probabilities, and then using the Moore-Penrose algorithm to solve the linear equation systems formulated by the class-wise averaged predicted probabilities and the bias gradients.

[1] Sun, Jingwei, et al. "Soteria: Provable defense against privacy leakage in federated learning from representation perspective." Proceedings of the IEEE/CVF conference on computer vision and pattern recognition. 2021.

**Summary Of The Review:**

Overall, this is a well-written paper which I'm leaning towards accepting. The authors' proposed method makes intuitive sense and the motivation is clear, the experiments are comprehensive. The score will be increased upon the satisfactory and convincing rebuttal regarding the questions in “Cons”.

---

> ### Author Response · Authors · 2022-11-12
> **Response to reviewer CT8P**
>
> Thanks for great appreciation of our work, such as "well-written", "high quality," and "considerable novelties", etc. As a new researcher, it is a great honor to receive these compliments. I will do my best to answer your remaining concerns.
>
> > **Q1: How the method proposed in this work addresses the issue that existing work cannot handle when the activation function yields negative values.**
>
> **R1:** Thanks for your comments. Since our approach utilizes only the last fully connected layer, **it does not matter which activation function is used for the previous network layers**. The reason why the prior methods [1,2] require non-negative activation is that **they expect to pass the unique negative sign for $\nabla z_i$ at the ground truth index $c$ $(\nabla z_i = p_i - y_i)$ to $\nabla \boldsymbol W_i$ that the adversary can obtain**, where $\nabla z_i$ denotes the derivative of the cross-entropy loss $\mathcal{L}$ w.r.t. the network final logit $z$ at index $i$, and the same goes for $\nabla \boldsymbol W_i$. According to the chain rule, we have $\nabla \boldsymbol W_i = \nabla z_i \boldsymbol e^{\top}$. As a result, **the above transitivity holds true only if the layer's input e is non-negative**. As for our method, it is **only related to the numerical magnitude of the gradient, but not to the gradient sign**.
>
> [1] Bo Zhao, Konda Reddy Mopuri, and Hakan Bilen. idlg: Improved deep leakage from gradients. arXiv preprint arXiv:2001.02610, 2020.
>
> [2] Hongxu Yin, Arun Mallya, Arash Vahdat, Jose M Alvarez, Jan Kautz, and Pavlo Molchanov. See through gradients: Image batch recovery via gradinversion. In Proceedings of the IEEE/CVF Conference on Computer Vision and Pattern Recognition, pp. 16337–16346, 2021.
>
> > **Q2: Is the reconstructed class-wise embedding reduced to the embedding produced by [3] when there is no bias for the FC layer? Ablation studies to compare the quality of class-wise embeddings learned by the methods of [1] and the proposed method’s.**
>
> Thank you for your suggestions. First of all, your thought is correct. **The bias gradient is an important component in our approach, and for a fully-connected layer without bias, we can indeed only obtain linear scaling of the embedding as in [3]**. Second, we performed an ablation experiment based on your suggestion and the results are as follows. (We execute attacks on CIFAR100 with the VGG-16 model at different training stages, the BatchSize is 64 and all results in the following table are the average of 20 replicate experiments.)
>
> Here (-) indicates that it was originally a negative value. As shown in the table, the cosine similarity of the embeddings recovered by our method at different training stages is greater than that of Soteria[3]. **This suffices to demonstrate that our restorations are of high quality**. Moreover, we note that our restored embeddings is more precise than Soteria's, but it doesn't hold up well after training. As in our R3(2) and R3(4) to Reviewer ngm7, **Approx 2 can be bypassed for the untrained model, while the entanglement leads to a non-negligible error of the embeddings in some cases after training**. In fact, in most cases, the MSE produced by our method is also less than Soteria's. The large MSEs in rare cases may lead to a large average MSE.
>
> | Training Stage | Metrics |  Soteria [3] | Ours |
> | :-------------| :------- | :------- | :------- |
> | Untrained (Epoch 0) | MSE   | 2.6e-4 | **1.6e-5** |
> | Untrained (Epoch 0)  | CosSim   | **0.978(-)** | **0.978** |
> | Middle (Epoch 40)  | MSE  | **2.0e-2** | 1.8e-1 |
> | Middle (Epoch 40)   | CosSim | 0.599(-) | **0.739** |
> | Trained (Epoch 100)  | MSE | **1.9e-1** | 5.3e-1 |
> | Trained (Epoch 100)   | CosSim | 0.396(-) | **0.734** |
>
> [3] Jingwei Sun, Ang Li, Binghui Wang, Huanrui Yang, Hai Li, and Yiran Chen. Soteria: Provable defense against privacy leakage in federated learning from representation perspective. In Proceedings of the IEEE/CVF conference on computer vision and pattern recognition, pp. 9311–9319, 2021.
>
> > **Q3: After knowing the number of instances per class within a batch, how do we assign the labels to each individual instance? Or how only knowing the count of each class within a batch can help recover features?**
>
> Thanks for your question! Since the adversary randomly initializes the dummy inputs of a batch as the optimization objectives in the gradient inversion attack, **which is originally unordered, we can assign any label to each instance according to the instance-wise label restoration results and they will eventually produce exactly the same batch-averaged gradient**. Please refer to R1&2 to Reviewer ngm7 for a detailed explanation of this issue.

---

> > ### Author Response · Authors · 2022-11-12
> > **Response to reviewer CT8P (Part 2)**
> >
> > > **Q4: When there are some basic defensive strategies applied (such as DP noise and gradient pruning), what are their influence on the proposed method?**
> >
> > Thanks for your comment. **Our method relies on the numerical accuracy of the gradients, so the noise introduced to them by a strong defense strategy will have a great impact on the attack**. We execute attacks on CIFAR100 with the ResNet-18 model at different training stages (untrained and well-trained), the BatchSize is 24 and all results in the following table are the average of 20 replicate experiments.
> > The two defense strategies, i.e., Differential Privacy (DP) and Gradient Sparsification/Pruning (GS/GP), are considered.
> >
> > As shown in the table, when the noise is introduced, **both LeAcc and LnAcc will decrease but not by much for the untrained model**. Furthermore, **for the trained model, moderate GS even improves performance a little**. We speculate that **the gradients with smaller magnitudes that are pruned are often insignificant and may reduce the errors generated by using them to calculate for restoring labels**.
> >
> > | Training Stage | Defense Method | Paramater Setting | LeAcc | LnAcc |
> > | :-------------| :------- | :----- | :---- | :----  |
> > | Untrained (Epoch 0)  | No Defense  | N/A | **1.000** | **1.000**  |
> > | Untrained (Epoch 0)  | Differential Privacy | Std 1e-4 | **1.000** | **1.000**  |
> > | Untrained (Epoch 0)  | Differential Privacy | Std 1e-3 | **1.000** | **1.000**  |
> > | Untrained (Epoch 0)  | Differential Privacy | Std 1e-2 | 0.980 | 0.973  |
> > | Untrained (Epoch 0)  | Gradient Sparsification | Ratio 10% | 0.990 | 0.989  |
> > | Untrained (Epoch 0)  | Gradient Sparsification | Ratio 20% | 0.990 | 0.989  |
> > | Untrained (Epoch 0)  | Gradient Sparsification | Ratio 40% | 0.986 | 0.984  |
> > | Untrained (Epoch 0)  | Gradient Sparsification | Ratio 80% | 0.960 | 0.957 |
> > | Trained (Epoch 100)  | No Defense | N/A | 0.845 | **0.817**  |
> > | Trained (Epoch 100)  | Differential Privacy  | Std 1e-4 | 0.822 | 0.798  |
> > | Trained (Epoch 100)  | Differential Privacy | Std 1e-3 | 0.778 | 0.768  |
> > | Trained (Epoch 100)  | Differential Privacy | Std 1e-2 | 0.706 | 0.702  |
> > | Trained (Epoch 100)  | Gradient Sparsification | Ratio 10% | 0.847 | 0.814 |
> > | Trained (Epoch 100)  | Gradient Sparsification | Ratio 20% | 0.852 | 0.814 |
> > | Trained (Epoch 100)  | Gradient Sparsification | Ratio 40% | **0.854** | **0.817**  |
> > | Trained (Epoch 100)  | Gradient Sparsification | Ratio 80% | 0.848 | 0.803 |

---

> > > ### Comment · Reviewer_CT8P · 2022-11-15
> > > **Very good rebuttal**
> > >
> > > Thanks for authors' very thorough and convincing rebuttal, most of my concerns are well-addressed. After reading the rebuttal, I believe that the quality, novelty, and empirical significance of this paper have reached the acceptance threshold of ICLR. Hence I will adjust my score accordingly.
> > >
> > > Follow up:
> > > Authors are encouraged to update their manuscript based on some problems pointed by the reviewers, for instance, I can see that my Q3 is also brought up by other reviewers, which suggests that this point is probably not clearly expressed in the currecnt version.

---

> > > > ### Author Response · Authors · 2022-11-15
> > > > **Sincere thanks to reviewer CT8P**
> > > >
> > > > Very glad to well-address most of your concerns, and thanks for your encouragement and raising the score! Morever, following your suggestions  and those of two other reviewers, I am working hard to further improve the manuscript as soon as possible.

---

### Official Review · Reviewer_ngm7 · 2022-10-25

**Confidence:** 3
**Correctness:** 3
**Technical Novelty And Significance:** 2
**Empirical Novelty And Significance:** 2
**Recommendation:** 6

**Clarity, Quality, Novelty And Reproducibility:**

This is an interesting work with originality. But the there are still some concerns of mine which limits the novelty. Please refer to the previous comments.

**Strength And Weaknesses:**

Strength:
1. The proposed method is compatible with gradient inversion attacks and can be plugged in to enhance current optimization-based methods.

2. Instead of optimization-based methods, this paper utilizes a method based on a pseudo-inverse algorithm.

Weakness:
1. Since the instance-wise label is the number of instances per class, how can this number be used with a model inversion attack? For example, IG attack assumes there is only one image per class in a batch, so it seems that with this attack, there is no large difference between previous class-wise label restoration and instance-wise label restoration.

2. Continued with point 1, what is the improvement of instance-wise restoration compared with class-wise restoration? Could you please give some application scenarios where this method surpasses the class-wise one?

3. I do not quite understand some of the approximations or assumptions in this paper as follows.
  (1) In the last of the Approx 1, you mention that "it is observed that the variance of \|e\| over the whole batch is so small". Could you please provide any theoretical analysis or any empirical results to explain this?
  (2) Approx 2 is based on the assumption that the gradients in $\mathbb{B}_i$ is dominant in the approximation of the gradients at index $i$. But what is the influence of label distribution in a batch to the restoration results? What if the number of class $i$ is much smaller than other classes (i.e., an extremely unbalanced batch)? Are there any results for this?
  (3) Approx 1 assumes the embeddings $e$ and the gradients of $L$ w.r.t. $z$ are similar over a certain class. The Approx 2 assumes the gradients in $\mathbb{B}_i$ is dominant. It is easier to understand this in a well-trained model. But what is the performance in the early training stage? Or could you please figure out whether the model used in the experiments is pre-trained or randomly initialized?
  (4) Could you please explain more about the Approx 3 of softmax? Why or when does this approximation hold?

Minor comments:
1. Superscript $j$ is used in Approx 1 without explanation, although it is explained in section 3.2.

2. In section 3.2, I think $k_j$ is the same as $|B_j|$. Is it correct?

**Summary Of The Paper:**

Gradient inversion in federated learning is often constrained by the lack of labels. Current label restoration methods are limited to class-wise label restoration, which is to infer the presence of a category. This paper introduces a new method to infer instance-wise labels in federated learning, i.e., the number of instances per class. Based on several approximations, the labels can be restored by solving a linear equation system using a pseudo-inverse algorithm. The authors then empirically explore how well this works in practice. Some results of the compatibility with model inversion attacks are also presented.

**Summary Of The Review:**

This is an interesting paper, but from my point of view, it relies on many assumptions to support the approximation and method proposed in this paper. There is a lack of evidence for these assumptions. Additionally, the improvement of this method compared with class-wise restoration seems limited. If the author can provide some motivation or results of the previously mentioned assumptions, I will raise my score.

---

> ### Author Response · Authors · 2022-11-11
> **Response to reviewer ngm7**
>
> We appreciate your praise for the originality and novelty of our paper and some concerns about the support of the theoretical assumptions. We give point-by-point responses below. We are pleased to offer more explanation or revisions if you have any other recommendations.
>
> >**Q1&2: How can the number of instances per class (the instance-wise labels) be used with a model inversion attack? And what is the improvement of instance-wise restoration compared with class-wise restoration?**
>
> **R1&2:** Thanks for your comments. The first issue is related to the gradient matching strategy, which is the foundation of the gradient inversion attack. In this strategy, the adversary, i.e., the honest-but-curious server, initializes a random batch of dummy inputs (with labels) as optimization objectives, which are originally disordered. **And no matter how we shuffle a training batch, the resulting batch-averaged gradient is consistent**. Therefore, **for each category with at least one instance, we simply select a subset with the capacity of the number of instances of this class from the remaining randomly initialized dummy inputs and assign them labels of this category**.
>
> Second, as you mentioned, the IG attack assumes only one instance per class in a batch, which is quite limited and unrealistic. We consider **a more real-world setting where there are multiple instances of a class, and previous methods cannot handle this situation**. **This is exactly the improvement from our instance-wise label restoration**. All of our experiments in Section 4 are carried out under the more reasonable scenario. As in Section 4.3, for a batch on MNIST contains at least 2 samples of class '0', class-wise label restoration prompts at most one random dummy input to optimize towards '0', while the rest can still only be unstably optimized with equally random labels.
>
> > **Minor Comments: (1) Superscript $j$ is used in Approx 1 without explanation, although it is explained in section 3.2; (2) In Section 3.2, is $k_j$ is the same as $|\mathbb B_j|$ ?**
>
> **R:** Thanks for your careful observation and reminder. As suggested, I will add the explanation of the superscript $j$ to Approx 1 in the future versions. In addition, your thought that $k_j$ is the same as $|\mathbb B_j|$ is correct, and they both represent the number of $j$-class instances.

---

> > ### Author Response · Authors · 2022-11-11
> > **Response to reviewer ngm7 (Part 2)**
> >
> > > **Q3: (1) Some theoretical analysis to explain that "it is observed that the variance of $||e||$ over the whole batch is so small"  in the last of the Approx 1.**
> >
> > **R3(1):** First of all, it is used to explain the low entanglement of weight gradients in  Approx 2. And the expression "the variance of $||e||$ over the whole batch is so small" is not so rigorous; In fact, it would be more appropriate to express it as "the variance of $||e||$ over the whole batch is **much smaller than the proportionality of the bias gradient**". For the untrained model, the original statement holds, **i.e., the embeddings in the entire batch are concentrative, for the rather poor classification ability at the beginning**; For the trained model, although the different categories of embeddings are separated, but the variance of $||e||$ is still not too large **due to normalized inputs and the commonly used BatchNormalization operation**.

---

> > > ### Author Response · Authors · 2022-11-12
> > > **Response to reviewer ngm7 (Part 3)**
> > >
> > > > **Q3: (2) Approx 2 is based on the assumption that the gradients in $\mathbb B_i$ is dominant in the approximation of the gradients at index $i$. But what is the influence of label distribution in a batch to the restoration results? What if the number of class $i$ is much smaller than other classes (i.e., an extremely unbalanced batch)? Are there any results for this?**
> > >
> > > **R3(2):** **The inter-class entanglement is significantly influenced by the label distribution**.  As in the extremely unbalanced case you presented, **the gradients of a minor category $i$ will be entangled with categories that consist of significantly more instances than it**, i.e., the dominance of $\mathbb B_i$ over the gradients at index $i$ is weakened.
> > >
> > > To prove this statement, we execute attacks on CIFAR100 with both untrained and trained VGG-16 models in batches ranging from 24 to 648. The results in the following tables are the average recovered *NoI* (*Number of Instances*) of 20 replicate experiments. We simply select 3 classes with class-id 0,18 and 92, whose numbers of instances are 1, BS-2 and 1 respectively (BS denotes the batch size and here is an unbalanced distribution).
> > >
> > > As shown in the following table, it can be seen that **for the trained model, when the degree of imbalance exceeds a certain threshold, the recovery effect for minor classes 0 and 92 deteriorates significantly**. However, for the untrained model,  our method can restore labels of minor classes perfectly even for such a disparate data distribution as 626:1. This is actually because in this case, **the errors arising from entanglement can be bypassed**. The detailed theoretical explanation is as follows:
> > >
> > > We reconstruct the class-wise embeddings based on $\\overline {\\boldsymbol{e}}_{\\mathbb B_i} \\approx {\\frac{\\overline{\\partial \\mathcal{L}}} {\\partial b_i}} _{\\mathbb B_i}^{-1} \\times {\\frac{\\overline {\\partial \\mathcal{L}}} {\\partial  \\boldsymbol W_i}} _{\\mathbb B_i}^{\\top} \\approx {\\frac{\\overline{\\partial \\mathcal{L}}} {\\partial b_i}}_\\mathbb B^{-1} \\times {\\frac{\\overline {\\partial \\mathcal{L}}} {\\partial \\boldsymbol W_i}}_\\mathbb B^{\\top}$. In Approx 2, if the error terms for weight and bias gradients are not ignored, we have:
> > >
> > > $\\quad \\quad \\quad \\quad {{\\frac{\\overline {\\partial \\mathcal{L}}} {\\partial b_i}}_\\mathbb B} = {\\frac{1}{|\\mathbb B|}} [{|\\mathbb B_i| {\\frac{\\overline {\\partial \\mathcal{L}}} {\\partial b_i}} _{\\mathbb B_i}} + \\sum_j  (|\\mathbb B_j| {\\frac{\\overline {\\partial \\mathcal{L}}} {\\partial b_i}} _{\\mathbb B_j})], j \neq i$, (1)
> > >
> > > $\\quad \\quad \\quad \\quad {{\\frac{\\overline {\\partial \\mathcal{L}}} {\\partial \\boldsymbol W_i}}_\\mathbb B} = {\\frac{1}{|\\mathbb B|}} [{|\\mathbb B_i| {\\frac{\\overline {\\partial \\mathcal{L}}} {\\partial \\boldsymbol W_i}} _{\\mathbb B_i}} + \\sum_j  (|\\mathbb B_j| {\\frac{\\overline {\\partial \\mathcal{L}}} {\\partial \\boldsymbol W_i}} _{\\mathbb B_j})], j \neq i$, (2).
> > >
> > > According to Approx 1,  ${{\\frac{\\overline {\\partial \\mathcal{L}}} {\\partial \\boldsymbol W_i}}_{\\mathbb B_i}}$ is approximately equal to
> > > ${{\\frac{\\overline {\\partial \\mathcal{L}}} {\\partial b_i}} _{\\mathbb B_i}}$ times the average $\\boldsymbol e^{\\top}$ over ${\\mathbb B_i}$. **For the untrained model, the average $\\boldsymbol e^{\\top}$ over any ${\\mathbb B_i}$ is almost equal**. Therefore, we have:
> > >
> > > $\\quad \\quad \\quad \\quad {{\\frac{\\overline {\\partial \\mathcal{L}}} {\\partial \\boldsymbol W_i}}_\\mathbb B} \\approx {\\frac{\\bar{\\boldsymbol e}^{\\top}}{|\\mathbb B|}} [{|\\mathbb B_i| {\\frac{\\overline {\\partial \\mathcal{L}}} {\\partial b_i}} _{\\mathbb B_i}} + \\sum_j  (|\\mathbb B_j| {\\frac{\\overline {\\partial \\mathcal{L}}} {\\partial b_i}} _{\\mathbb B_j})], j \neq i$, (3).
> > >
> > > **Since the restored embedding is the quotient of the two formula (3) and (1), the error terms do not affect the proportional result in this case**. A more vivid example is that:
> > >
> > > The error terms of the weight and bias gradient represent sugar and water respectively. **Due to their proportions are consistent with the original concentration of sugar water, adding them will not change the concentration**.
> > >
> > > |   BatchSize |    24  |    72   |  216   |   648  |
> > > | :-------------  | :------- | :------ | :------ | :------  |
> > > | Trained + Imbalanced + Cls0| **1** | **1** | 10 | 51 |
> > > | Trained + Imbalanced + Cls18| 18 | 57| 155 | 371 |
> > > | Trained + Imbalanced + Cls92| **1** | **1** | 9 | 65 |
> > > | Untrained + Imbalanced + Cls0| **1** | **1** | **1** |**1** |
> > > | Untrained + Imbalanced + Cls18| 22 | 70 | 214 | 646 |
> > > | Untrained + Imbalanced + Cls92|**1**| **1** | **1** | **1** |

---

> > > > ### Author Response · Authors · 2022-11-12
> > > > **Response to reviewer ngm7 (Part 4)**
> > > >
> > > > > **Q3: (3) What is the performance in the early training stage? Are the models used in the experiments pre-trained or randomly initialized?**
> > > >
> > > > **R3(3):** **The performance is quite excellent in the early training stage**, and our experiments in the paper are all carried out on the untrained (randomly initialized) model. We execute attacks on CIFAR100 with the VGG-16 model at different training stages, the BatchSize is 64 and all results in the following table are the average of 20 replicate experiments. Obviously, we can find that **the attack effect gets worse after model training**.
> > > >
> > > > As you said, the well-trained models seem to be more satisfied with intra-class similarity and inter-class entanglement. However, the result is exactly the opposite of the idea. In fact, **the approximations are not wrong, but our previous naming or description of these approximations is imprecise and somewhat misleading**. We modify them into ''*Intra-class Uniformity and Concentration of Embedding Distribution*'' and ''*Inter-class Low Entanglement of Gradient Contributions*''. The detailed explanation is as follows:
> > > > - **Intra-class Uniformity and Concentration of Embedding Distribution:**  Approx1 and 3 essentially replace the embeddings of a certain class with its arithmetic mean, i.e., the geometric center. **It holds when the embedding distribution is uniform and concentrated**.
> > > >     - Therefore, this property is more satisfied for the untrained model. If we project the embeddings on a 2D plane, its distribution is **close to a uniform circle**. Moreover, we can achieve as close to 100% accuracy as the proposed method by directly taking all probabilities as $1/n$ to restore the labels, where $n$ is the number of classes.
> > > >     -  For the trained model, although the various categories can be well separated, the internal density of a certain category may **not be sufficiently uniform and symmetrical, and there will be some outliers**.
> > > > - **Inter-class Low Entanglement of Gradient Contributions:** Approx2 indicateds that the $i$-class sample mainly contributes to the $i$-th gradient row. According to our derivation that $\nabla b_i = \nabla z_i = p_i - y_i $, the bias gradient of $i$-class sample at index $i$ is $p_i^i-1$, while that of another class $j$ is $p_i^j, j \neq i$ . Here the superscripts correspond to the categories. Therefore, we hope that $|p_i^j| \ll |p_i^i-1|$ for any $j \neq i$.
> > > >     - For the untrained model, all probabilities can be approximated by $1/n$. In this case, we consider whether $|1/n-1| \ll |1/n|$ holds. **It is not difficult to find that it is related to the label distribution**. The larger $n$ is, the easier it is to satisfy. In addition, **the label distribution should not be too disparate**. Otherwise, the smaller $p_i^j$ accumulates too much and cannot be ignored any more. Fortunately, **Approx2 can be bypassed when restoring the embeddings as explained in R3(2)**.
> > > >     - For the trained model, **the magnitude of the gradients will obviously be greatly reduced**.  Therefore, **the relative error can easily become larger**. For instance, if we take  $p_i^i$ as 0.99 and $p_i^j$ as 0.01, then $|p_i^i-1| = |0.99-1| = 0.01$ and thus $|p_i^j| \ll |p_i^i-1|$ no longer holds. When class $j$ is similar to class $j$, the entanglement will be greater.
> > > >
> > > > |   Epochs |    Training Acc  |   LeAcc   |  LnAcc   |
> > > > | :------ | :------- | :------ | :------ |
> > > > | 0 | 0.002 | **1.000** | **1.000** |
> > > > |10 | 0.447 | 0.805 | 0.722 |
> > > > |20 | 0.533 | 0.785 | 0.710|
> > > > |30|  0.565 | 0.781 | 0.691|
> > > > |40 | 0.581 | 0.811 | 0.722 |
> > > > |50 | 0.816 | 0.676 | 0.593 |
> > > > |60 | 0.880 | 0.673 | 0.589 |
> > > > |70 | 0.897 | 0.685 | 0.608 |
> > > > |80 | 0.979 | 0.647 | 0.570 |
> > > > |90 | 0.989 | 0.675 | 0.579 |
> > > > |100 | 0.994 | 0.656 | 0.567 |
> > > >
> > > > > **Q3: (4) Explain more about the Approx 3 of softmax? Why or when does this approximation hold?**
> > > >
> > > > **R3(4):**  Approx 3 also relies on the intra-class similarity [or a more reasonable description mentioned in R3(3)——intra-class uniformity and concentration of embedding distribution], and **thus applies to both untrained and trained models (slightly worse than the untrained model)**. In addition, the approximation is actually equivalent to determining whether $f(x+y)$ is equal to $f(x)+f(y)$. The linear mapping by the last layer is obviously satisfied, so **the error primarily comes from the softmax transform**. And the softmax transformation has a property of translation invariance. **If the errors of each component of the output are approximately the same, the Approx 3 can also be considered to hold**.

---

> > > > > ### Author Response · Authors · 2022-11-12
> > > > > **Response to reviewer ngm7 [Error analysis for the experiment in R3(3)]**
> > > > >
> > > > > For the experiment in R3(3), we conduct error analysis shown in the table below. We choose the models
> > > > > at epoch 0, 40 and 100 respectively as the representatives of the three stages of untrained, mid-training and well-trained. The three approximations and restored embeddings are our check items. And the evaluation metrics include MSE (Mean Squared Error), MRE (Mean Relative Error) and CosSim (Cosine Similarity), where MRE is a proportional value.
> > > > >
> > > > > First of all, both Approx1 and Approx3 are related to *Intra-class Uniformity and Concentration of Embedding Distribution*. **Their MSE results are consistent with this property: best at the beginning, second at the end, and worst in the middle.** However, since the gradient magnitude becomes smaller as the training progresses, the mid-term MRE may be less than the final (for Approx1 here).
> > > > >
> > > > > Secondly, Approx2 represents *Inter-class Low Entanglement of Gradient Contributions*. Due to the complexity of the error of the weight gradient (Approx2-2), we choose to analyze the bias gradient (Approx2-1). **As the training progresses, MSE decreases but MRE increases, indicating that the entanglement is actually increasing.** This also agrees with the explanation in R3(3).
> > > > >
> > > > > Finally, we note that the recovered embeddings for untrained model are the best. **This is because it satisfies both properties and can even bypass Approx2**. In addition, we can also assert from the above results that **the error of Approx 2 may be larger than that of Approx1**.  And since Approx2 directly participates in the division calculation for restoring embeddings, which intuitively may also be a reason.
> > > > >
> > > > > | Check Item | Untrained (Epoch 0) | Middle (Epoch 40) | Trained (Epoch 100) |
> > > > > | :-------------| :------- | :------- | :------- |
> > > > > | Approx1 MSE | **1.6e-11** | 6.6e-5 | 2.8e-6 |
> > > > > | Approx1 MRE | **0.000** | 0.668  | 0.822 |
> > > > > | Approx2-1 MSE | 9.6e-5 | 5.7e-5  | **3.0e-7** |
> > > > > | Approx2-1 MRE | **1.377** | 3.615  | 13.218 |
> > > > > | Approx2-2 MSE | **2.4e-8** | 6.5e-7  | 4.0e-8 |
> > > > > | Approx2-2 MRE | 2.896 | **1.391**  | 6.129 |
> > > > > | Approx3 MSE | **6.1e-16** | 9.1e-5  | 2.6e-6 |
> > > > > | Approx3 MRE | **0.000** | 0.157  | 0.130 |
> > > > > | Embedding MSE | **1.6e-5** | 1.8e-1  | 5.3e-1 |
> > > > > | Embedding CosSim| **0.978** | 0.739  | 0.734 |

---

> ### Author Response · Authors · 2022-11-22
> **Looking forward to your feedback**
>
> Dear reviewer ngm7:
>
> Thanks again for your constructive and valuable comments, which have helped us improve the paper considerably. We believe that your concerns about theoretical assumptions have been addressed somewhat satisfactorily. Sincerely hope to gain further valuable feedback from you and continue to improve this work. We are also willing to clarify any additional concerns.
>
> Best regards, Authors of #1628

---

> > ### Comment · Reviewer_ngm7 · 2022-11-25
> > **Acknowledgement of Rebuttal**
> >
> > Thanks for the authors' comprehensive rebuttal. With sufficient and detailed experiments and analysis, most of my concerns have been addressed. So I am raising my score.

---

> > > ### Author Response · Authors · 2022-11-26
> > > **Thanks for raising the score**
> > >
> > > Dear reviewer ngm7:
> > >
> > > We are very glad that our rebuttal has addressed most of your concerns, and sincerely thanks for your raising the score! Your highly valuable comments benefit us a lot.
> > >
> > > Best regards, Authors of #1628

---

### Author Response · Authors · 2022-11-11
**Response to all reviewers**

Thanks very much for all reviewers’ careful reading and valuable comments. We have benefited significantly from your expertise and are encouraged by such appreciations of our work. Due to space constraints, we have not previously explained the theoretical assumptions in detail. Therefore, we endeavor to individually address the concerns of each reviewer and sincerely hope that this response can help you better understand our work. A revision (including the manuscript and supplementary materials) has been released. If your concerns remain unresolved or new ones arise, I will be happy to clarify them for you.

---

> ### Author Response · Authors · 2022-11-24
> **Summary of the revision**
>
> Dear reviewers:
>
> We are pretty grateful for your careful and meticulous comments, which prompted us to improve the manuscript. To make it easier for you to check and understand, we summarize the changes as follows:
>
> * We have combined the original Figure 1 (Threat Model) and Figure 2 (Overview of The Method) into one, and simplified the painting to make it more intuitive.
>
> *  As mentioned in the R3(3) to reviewer ngm7, we have **modified the names of the two empirical attributes** on which the embedding reconstruction is based from *Intra-class High Similarity* & *Inter-class Low Entanglement* to ***Intra-class Uniformity and Concentration of Embedding Distribution & Inter-class Low Entanglement of Gradient Contributions*** (**in Section 3.1**). This more accurately describes our approximations and avoids being misleading. Moreover, a more detailed explanation is also provided (**about R3, for instance, when and why the 3 approximations are satisfied**).
>
> * We have added a few experiments and analysis for your concerns.
>
>     * One is about **the effect of different training stages** from reviewer ngm7 (an experiment in Section 4.1 and **its corresponding error analysis** in Appendix D). We have demonstrated that **the attack against the untrained model is the most effective**, and clarified that **the experiments in the original paper were all conducted under this setting**;
>     * **The impact of extreme label distribution on attacks** is discussed in Appendix F and we further introduce a finding that **Approx 2 can be bypassed for untrained models** in Section 3.1;
>     * Following the suggestions of reviewer CT8P, we have **compared the quality of class-wise embeddings at different training stages by Soterta[1] and the proposed method’s  in Appendix E** and **discussed two typical defense schemes in Appendix G**.
>
> *  We have **supplemented the description of how instance-wise labels can be used to enhance existing gradient inversion attacks in Section 4.4**, which is a concern of both reviewers ngm7 and CT8P.
>
> Best Wish,
> Authors
>
> [1] Jingwei Sun, Ang Li, Binghui Wang, Huanrui Yang, Hai Li, and Yiran Chen. Soteria: Provable defense against privacy leakage in federated learning from representation perspective. In Proceedings of the IEEE/CVF conference on computer vision and pattern recognition, pp. 9311–9319, 2021.

---

### Decision · Program_Chairs · 2023-01-20

**Decision:**

Accept: poster

**Justification For Why Not Higher Score:**

It still lacks a strong theoretical results.

**Justification For Why Not Lower Score:**

This paper is novel and is strongly supported by experiments. The work does not need strong assumptions for recovering labels and the empirical results are state-of-the-art.

**Metareview: Summary, Strengths And Weaknesses:**

This paper studies instance-wise batch label restoration in the realm of federated learning. It proposes an analytic method which requires merely the gradient of the final layer. This work implements Moore-Penrose pseudoinverse algorithm to estimate the Number of Instances (NoI) per class by solving a linear system. Reviewers think this paper is novel and is strongly supported by experiments. One reviewer thought that the work does not need strong assumptions for recovering labels and the empirical results are state-of-the-art. I think the paper will be of interest to Federated learning researchers. I suggest acceptance.


**Note From Pc:**

if the above contains the word "oral" or "spotlight" please see: "oral" presentation means -> notable-top-5% and "spotlight" means -> notable-top-25%. As stated in our emails, we are disassociating presentation type from AC recommendations